# Development of a Phytomelatonin-Rich Extract from Cultured Plants with Excellent Biochemical and Functional Properties as an Alternative to Synthetic Melatonin

**DOI:** 10.3390/antiox9020158

**Published:** 2020-02-16

**Authors:** Francisca Pérez-Llamas, Josefa Hernández-Ruiz, Alberto Cuesta, Salvador Zamora, Marino B. Arnao

**Affiliations:** 1Department of Physiology, Faculty of Biology, University of Murcia, 30100 Murcia, Spain; frapella@um.es (F.P.-L.); sazana@um.es (S.Z.); 2Department of Plant Biology, (Plant Physiology), Faculty of Biology, University of Murcia, 30100 Murcia, Spain; jhruiz@um.es; 3Department of Cell Biology & Histology, Faculty of Biology, University of Murcia, 30100 Murcia, Spain; alcuesta@um.es

**Keywords:** food supplement, jet-lag, melatonin, nutraceutical, phytomelatonin, sleep disorders

## Abstract

Melatonin is a pleiotropic molecule with multiple and various functions. In recent years, there has been a considerable increase in the consumption of melatonin supplements for reasons other than those related with sleep (as an antioxidant, for anti-aging, and as a hunger regulator). Although the chemical synthesis of melatonin has recently been improved, several unwanted by-products of the chemical reactions involved occur as contaminants. Phytomelatonin, melatonin of plant origin, was discovered in several plants in 1995, and the possibility of using raw plant material as a source to obtain dietary supplements rich in phytomelatonin instead of synthetic melatonin, with its corresponding chemical by-products was raised. This work characterizes the phytomelatonin-rich extract obtained from selected plant material and determines the contents in phytomelatonin, phenols, flavonoids, and carotenoids. Additionally, the antioxidant activity was measured. Finally, a melatonin-specific bioassay in fish was carried out to demonstrate the excellent biological properties of the natural phytomelatonin-rich extract obtained.

## 1. Introduction

Melatonin (*N*-acety-5-methoxyltryptamine) is a biogenic indoleamine with an extensive range of cellular and physiological actions. It was discovery in bovines in 1958 [1] and in 1959 in humans [2,3]. The first related action of melatonin was that it enlightens skin color changes in fish, frogs, and tadpoles as a consequence of the aggregation of the melanophores present in melanocytes, which is why the substance was called melatonin [1,4]. Today, melatonin is related with a broad spectrum of functions, among which the actions influencing circadian rhythms, mood, sleep, body temperature and activity, food intake, endocrine rhythms, sexual behavior, and neuronal and immunomodulatory activity have been the most studied [5,6,7,8,9,10,11,12,13]. More recently, interesting studies on the role of melatonin regulating metabolic pathways have been published [14,15,16]. For example, melatonin regulates glucose metabolism, inducing nocturnal insulin resistance and diurnal insulin sensitivity. This effect seems to be associated with nocturnal fasting and diurnal feeding, preventing excessive body weight gain [17,18,19]. Additionally, some interesting studies on the oncostatic properties of melatonin in in vivo and in vitro experimental models have appeared [12,20,21,22]. Thus, the roles of melatonin in many tumors, both in growth, proliferation, apoptosis, and metastasis have been demonstrated [23,24]. The treatments of melatonin with chemo- and radiotherapy improves the sensitivity of tumors to nonproliferation by drugs, acting as a synergistic molecule in the control of cancer cells. Additionally, melatonin mitigates acute damage in normal cells, protecting them against drug toxicity, possibly enhancing immunomodulation [25,26,27,28]. More recently, the use of electrophile-melatonin derivatives as chemopreventive agents on cancer cells has been demonstrated, calling them new “intelligent” molecules [29]. However, without a doubt, the most known and well-studied roles of melatonin are those related with sleep regulation and circadian rhythms. In this sense, melatonin is produced by the pineal gland in the night, increasing blood melatonin levels during the first period of sleep to around 150–220 pmoles/mL, acting on sleep initiation, reducing sleep latency and fragmentation and increasing sleep duration and sleep quality [30,31,32]. Additionally, melatonin, being a powerful chronobiological molecule that contributes to the correct circadian behavior of physiological functions, acts as an internal synchronizer of the circadian sleep-wake cycle and seasonal rhythmicity. In this sense, many sleep disorders have been treated with melatonin, including delayed sleep phase syndrome, night shift-work sleep disorder, seasonal affective disorder, sleep disorders in the blind and aging, and patho-physiological disorders of children, with notable improvements in sleep quality [9,32,33,34,35,36]. The most widespread disorder treated with melatonin is jet-lag, a de-phasing in the sleep-wake rhythms following trans-oceanic flights [37,38,39].

The regulation of melatonin supplements differs between countries and areas. Thus, dietary supplements is the term used by the American Food and Drug Administration (FDA) [40,41], while in the European Community (EC) the same are defined as food supplements by the European Nutraceutical Association (ENA) [42,43]. An important difference is while dietary supplements as defined by the FDA are not considered drugs and are not subject to control or evaluation, the ENA classifies and controls food supplements. With respect to melatonin supplements, the melatonin dosage is not limited in the USA, while EC regulations require melatonin supplements to contain less than 2 mg/dose or day. At present, commercial melatonin supplements contain exclusively synthetic melatonin and by-products such as 1,2,3,4-tetrahydro-β-carboline-3-carboxylic acid, 3-(phenylamino)-alanine (PAA), 1,1-ethylidenebis-(tryptophan) (so-called *peak E*), among others (Table 1) are among the most common compounds which may appear in synthetic melatonin preparations [44,45]. These by-products, generally tryptophan-related compounds, including “*peak E*”, which is related to Eosinophilia Myalgia Syndrome (EMS), an incurable and fatal disease which, in 1993, killed 37 people and affected 1511 in the USA, due to a defective criteria for the manufacture of tryptophan, the precursor of melatonin during its synthesis [46,47,48]. PAA and *peak E* were also related with the Spanish rapeseed oil syndrome in 1981, which killed around 900 people and left another 20,000 affected [48,49]. In other cases, melatonin was generated from phthalimide, in a very fast, easy, and cheap chemical synthesis involving only four steps. In this case, up to 14 contaminants have been described, requiring toxic substrates or catalysts such as pyridine, triethylbenzenylamine chloride, 1,3-dibromopropane and *p*-anisidine [50,51,52,53]. However, some modern methods of the chemical synthesis of tryptamine derivatives, including melatonin, have reduced the appearance of many of the by-products [54,55].

Phytomelatonin is the term used for melatonin of plant origin. Phytomelatonin was discovered in plants in 1995 [56,57] and has been detected practically in all the plant species analyzed to date, with a high variability in content, depending on the plant species and the plant organ or tissue in question [58]. In general, aromatic and medicinal plants seem to have the highest phytomelatonin contents [59]. Many physiological studies have been looked at the role of phytomelatonin in plants, where it has been found act as a protective molecule against physical, chemical, or biological stressors [60,61,62,63,64,65]. In general, phytomelatonin seems to be involved in many gene regulation processes related with metabolism, growth and development and the actions of plant hormones [66,67,68,69,70,71,72]. Additionally, its biosynthesis pathway in plants has been well characterized, and seen to differ in some steps with respect to the melatonin biosynthesis pathway in animal cells [73,74].

Although in melatonin preparations as pharmaceuticals (drugs), purity is high and, therefore, the occurrence of the above mentioned by-products is unlikely, this is not the case of most dietary or food supplements containing synthetic melatonin. However, food supplements of plant origin tend to be more accepted than others by consumers. In addition, the increase in consumption of melatonin supplements in recent years for reasons that go beyond the specifications related to sleep (e.g., as an antioxidant, in anti-aging, as a hunger regulator, etc.), makes it interesting to bet on active materials of natural origin, in addition to raising quality levels and their controls [75]. This has given rise to “green” objectives for obtaining dietary supplements rich in phytomelatonin rather than synthetic melatonin. In this work, a phytomelatonin-rich extract obtained from selected plant material is presented. This is possibly the first extract rich in phytomelatonin that is obtained with the possibility of being commercialized. Its richness in phytomelatonin, phenols, flavonoids, antioxidant activity, and carotenoids is analyzed. Additionally, a melatonin-specific bioassay using fish is described, pointing to the excellent biological properties of the natural phytomelatonin-rich extract obtained.

## 2. Materials and Methods

### 2.1. Chemicals and Reagents

All the chemicals and standard reagents used were from Sigma-Aldrich (Madrid, Spain). Milli-Q system (MilliPore-Sigma Corp, Darmstadt, Germany) ultra-pure water was used.

### 2.2. Plant Material

A mixture of diverse plants was used. The exact formulation of herbal mix (HM) and the protocol are being patented. HM were treated according the scheme in Figure 1. From HM, a concentrated final product so-called Bioriex was obtained. Both HM plants and Bioriex were analyzed to measure the biochemical parameters described below. 

### 2.3. Proximate Analysis

Samples of HM and Bioriex were analyzed for moisture (before drying, method no. 945.15), ash (method no. 942.05), crude protein (Kjeldahl method using a factor of 6.25, method no. 920.54), crude fat (method no. 920.39), and dietary fiber (method no. 985.29) contents, according to the AOAC methods [76].

### 2.4. Determination of Total Phenolic Content

Folin–Ciocalteu’s reagent [77] was used to determine the total content of phenolic compounds in the samples as in [78]. Five hundred µl of sample was placed in a glass test tube, and 0.85 mL of water, 50 µL of 1N NaOH, and 50 µL of Folin–Ciocalteu’s reagent were added. The reaction medium was allowed to react in the dark, at 30 °C, for 1 h, before the absorbance at 755 nm was measured with a UV–VIS spectrophotometer (Perkin-Elmer GmbH, model Lambda 2S, Rodgau, Germany). The results are expressed as moles of gallic acid (used as standard) equivalents per gram of dry matter.

### 2.5. Determination of Total Flavonoid Content

The aluminum chloride colorimetric method, modified by Woisky and Salatino (1998), was applied [78,79]. Quercetin was used as a standard. Quercetin (10 mg) was dissolved in 80% ethanol and then diluted to 12.5, 25, 50, 75, and 100 µg/mL. The diluted standard solutions were separately mixed with 50 µL of 10% aluminum chloride, 50 µL of 1 M potassium acetate and 0.85 mL of distilled water. After incubation at 30 °C for 30 min, the absorbance of the reaction mixture was measured at 415 nm. A similar procedure was applied to HM and Bioriex samples for total flavonoid content analysis. The results were expressed as moles of quercetin equivalents per gram of dry matter.

### 2.6. Determination of Hydrophilic Antioxidant Activity

Hydrophilic antioxidant activity was measured in the samples using the method described by Arnao et al. (1999) [80,81], which is based on the ability of the antioxidants of a sample to reduce the radical cation of 2,2′-azino-bis-3-(ethylbenzothiazoline-6-sulphonic acid) (ABTS^·+^), determined by the descoloration of ABTS^·+^ and measuring the quenching of the absorbance at 730 nm. This activity was calculated by comparing the values of the sample with a standard curve of ascorbic acid and expressed as moles of ascorbic acid equivalents per gram of dry matter.

### 2.7. Carotenoid Content

The carotenoid content was estimated using the method described by Lichtenhaler and Wellburn (1983) [82], in which HM and Bioriex samples (0.1 g) were extracted in 80% acetone The absorbance at 450 nm was measured in the extraction phase and expressed as moles of β-carotene equivalents per gram of dry matter.

### 2.8. Phytomelatonin Content Measurements

Samples for phytomelatonin analysis were prepared according to [61,83]. Briefly, HM or Bioriex (0.1 g) was placed in vials containing ethyl acetate (3 mL). After leaving overnight (15 h) at 4 °C in darkness with shaking, the tissue was discarded after being washed with the respective solvent (0.5 mL). The extract and washing solution from each sample were evaporated to dryness under vacuum using a SpeedVac (ThermoSavant SPD111V, Thermo-Fisher Sci, Waltham, MA, USA) coupled to a refrigerated RVT400 vapor trap. The residue was redissolved in acetonitrile (1 mL), filtered (0.2 μm), and analyzed using liquid chromatography (LC) with fluorescence detection and LC-QTOF/MS. The procedures were carried out in dim artificial light.

#### 2.8.1. Phytomelatonin Analysis by Liquid Chromatography with Fluorescence Detection

A Jasco liquid chromatograph Serie-2000 (Tokyo, Japan) equipped with an online degasser, binary pump, auto sampler, thermo stated column and a Jasco FP-2020-Plus fluorescence detector were used to measure phytomelatonin levels. An excitation wavelength of 280 nm and an emission wavelength of 350 nm were selected. A Waters Spherisorb-S5 ODS2 column (250 × 4.6 mm) was used. The isocratic mobile phase consisted of water:acetonitrile (80:20) at a flow rate of 0.2 mL/min. The data were analyzed using the Jasco ChromNAV v.1.09.03 Data System Software. For correct identification, an in-line fluorescence spectral analysis (using the Jasco Spectra Manager Software) compared the excitation and emission spectra of standard melatonin with the corresponding peak of phytomelatonin in the samples [83].

#### 2.8.2. Phytomelatonin Analysis by Liquid Chromatography with Time-of-Flight/Mass Spectrometry (LC-QTOF/MS)

Identification of phytomelatonin in plant extracts was also confirmed using a LC/MS system consisting of an Agilent 1290 Infinity II Series LC (Agilent Technologies, Santa Clara, CA, USA) equipped with an Automated Multisampler module and a High Speed Binary Pump, and connected to an Agilent 6550 Q-TOF Mass Spectrometer (Agilent Technologies, Santa Clara, CA, USA) using an Agilent Jet Stream Dual electrospray (AJS-Dual ESI) interface. Experimental parameters for HPLC and Q-TOF were set in MassHunter Workstation Data Acquisition software (Agilent Technologies, Rev. B.08.00) [84].

Samples (HM and Bioriex) were filtered through 0.2 µm filters before analyzing. Standards or samples (20 µL) were injected onto a Waters XBridge C18 5 µm, 100 × 2.1 mm LC column, at a flow rate of 0.4 mL/min, and thermo stated at 40 °C. Solvents A (MilliQ water with 0.1% formic acid) and B (acetonitrile with 0.1% formic acid) were used for the compound separation with an initial condition of 95% solvent A and 5% solvent B. After injection, the initial conditions were maintained for 2 min and then, compounds were eluted using a linear gradient 5–100% solvent B for 8 min. 100% solvent B was maintained for 2 min, and the system was finally equilibrated at 5% B for 3 min before a new injection.

The mass spectrometer was operated in the positive mode. The nebulizer gas pressure was set to 40 psi, and the drying gas flow was set to 13 L/min at a temperature of 250 °C. The sheath gas flow was set to 12 L/min at a temperature of 300 °C. The capillary spray, nozzle, fragmentor, and octopole 1 RF Vpp voltages were 3500 V, 50 V, 150 V, and 750 V, respectively. Profile data in the 50–300 *m*/*z* range were acquired for MS scans in 2 GHz extended dynamic range mode. A reference mass of 121.0509 was used. Data analysis was performed with MassHunter Qualitative Analysis Navigator software (Agilent Technologies, Rev. B.08.00). The signal corresponding to phytomelatonin was extracted and quantified with *m*/*z* of 233.1285.

### 2.9. Biological Assay in Fish: In Vivo Imaging of Fish Fin Melanophores

Two specimens of the teleost fish gilthead seabream (*Sparus aurata*) and European sea bass (*Dicentrarchus labrax*) of 5–10 g body weight were used. The fishes were maintained in laboratory aquaria with closed-recirculating marine systems (28‰ salinity, 20 ± 2 °C, 12 h light:12 h dark photoperiod) and sacrificed by an overdose of 40 ppm clove oil as in [85]. Animals used for this study were reused from other experiments (Bioethical Committee of the University of Murcia, approval A13150104) fulfilling the “3 Rs” principle. 

Caudal fins were excised and samples of 2–3 mm were placed onto a glass slide and examined under a Leitz Laborlux-12 light microscope (Wetzlar, Germany). A drop of 10 µL of synthetic melatonin at 1 µM (as positive control) or Bioriex samples (10 µL) were added and melanophores recorded with a video camera (Sony SSC-C370P, Sony Co., Tokyo, Japan) up to 20 min using PCTV Vision software (Pinnacle Systems, Corel Inc., Ottawa, Canada). Videos were further processed with VSDV Free video editor (Flash-Integro LLC, MultiLab LLC, Spring Lake, MI, USA) and representative images analyzed with ImageJ software [86].

### 2.10. Statistical Analysis

For all the data, differences were determined using the SPSS 10 program (SPSS Inc., Chicago, IL, USA), applying the LSD multiple range test to establish significant differences at *p* < 0.05. The results are expressed as mean ± standard error (SE, *n* = 5).

## 3. Results and Discussion

A mixture of herbs (HM) was used to obtain phytomelatonin-rich extracts, so-called Bioriex (Figure 1). The plant material, the elicitation and phytomelatonin extraction protocols are being patented. These plants proceeding from non-transgenic seeds were obtained by organic culture, free of pesticides and fertilizers. A green-extractive process was applied to obtain stabilized and natural phytomelatonin-rich extracts ready for use in several applications. 

Table 2 shows the results of a proximate analysis of the plant material (HM) and the final extract (Bioriex) with details of several components. Of particular note is the high protein, carbohydrate, and fiber content, and low fat content in HM. However, the significant quantitative difference in all the parameters between the plants (HM) and the final extract obtained (Bioriex) would be due to the intensive phytomelatonin concentration process applied. Thus, plant oils were the main component of Bioriex, where phytomelatonin is perfectly solubilized and extracted.

Figure 2 shows representative records of the respective chromatographic analyses of standard melatonin and of the concentrated extract Bioriex (see Figure 1). The chromatograms were recorded by the fluorescence detector and by the mass detector-QTOF. Both LC methods showed excellent quantitative parameters to the determination of phytomelatonin in complex plant extracts, according to previously published data [83,84].

Table 3 shows the phytomelatonin contents of the initial plant material (HM) and of the final concentrated product (Bioriex). As can be seen, there is a difference of three orders of magnitude. In the case of HM, the phytomelatonin content was multiplied by almost 1000 with respect to un-elicited plants (data not shown), meaning that significant amounts of phytomelatonin can be extracted and concentrated from HM. We are currently modifying some parameters that will allow us to reach even higher phytomelatonin levels in Bioriex than those shown. However, values of around 7 mg/g of phytomelatonin may be sufficient to be able to prepare nutritional supplements.

Generally, the phytomelatonin content of several plants varied considerably, the usual phytomelatonin content being around of 1–100 nanograms by gram of plant material (DW or FW) [58]. In the case of many medicinal and aromatic plants (MAP), higher phytomelatonin contents have been described [59]. For example, thyme (*Thymus vulgaris* L.), liquorice root (*Glycyrrhiza uralensis* Fisch.), sage (*Salvia officinalis* L.), St. John’s wort plant (*Hypericum perforatum* L.) showed maxima of 38, 34, 29, and 23 µg of phytomelatonin/g DW, respectively, all higher than the content of coffee beans (5–6 µg/g DW) [87]. Chinese MAPs have also been seen to have a high phytomelatonin content. Some common Mediterranean MAPs such as myrtle, laurel, laurestine, buckthorn and sage have a phytomelatonin content ranging from 0.3 to 8 ng/g FW, lowest content compared with the high content described for Chinese herbs [88]. Other well-known MAPs such as feverfew (*Tanacetum parthenium*) also have a high phytomelatonin content (1.5 µg/g DW) [89]. 

One of the peculiarities observed in studies of phytomelatonin has been the great variability of its endogenous content in plants. Phytomelatonin contents ranging from a few nanograms to quite considerable amounts, measurable in micrograms, have been recorded per gram of plant tissue. Generally, seeds, leaves, stems, seedlings and roots present the highest phytomelatonin levels and fruits the lowest. In general, MAPs have significantly higher levels of phytomelatonin than fruits. However, these are only rough estimates because, depending on the variety, growing conditions and other circumstances, the contents may vary. Nevertheless, the influence of harvest is a relevant factor that determines the final phytomelatonin contents in plants. This great variability in the data has created disbelief about the phytomelatonin content of some plants, an aspect also being influenced by geographical and seasonal components [58,90]. For these reasons, we have opted to generate a plant material rich in phytomelatonin under controlled conditions, avoiding the heterogeneity associated with many of the plant raw materials. Our process is more expensive and laborious but allows us to obtain concentrated extracts very rich in phytomelatonin, and is available for use as an active material in food supplements and other applications [91,92]. 

Table 3 also shows the values of other parameters of interest such as total phenolic content, total flavonoid content and hydrophilic antioxidant activity. Logically, HM showed lower values of antioxidant activity, total phenols and total flavonoids than Bioriex, following the concentrating process has undergone. The substantial amount of carotenoids detected conferred a pale yellow color to the Bioriex. The presence of all these compounds, among others in Bioriex, clearly stabilizes the final extract, and also contributes to stabilizing the phytomelatonin molecules. Nutritionally, these compounds (phenols, flavonoids, carotenoids) are healthy compounds, with multiple beneficial properties for human and animal health [93,94], which adds to the natural quality of the phytomelatonin-rich extracts (Bioriex).

The first response attributed to melatonin was the pigment aggregation response in amphibians and fish, which explains how the molecule received its name [1]. Melanophores are specialized pigmented cells present in the fish scales of many fish, and in the skin of amphibians and mammals. In fish and amphibians, melanophores allow fast color changes, mainly in response to stress environment conditions. When the melanosomes, containing pigmented granules, aggregate, the skin becomes lighter in color, whereas dispersal of the same pigments throughout the cell results in skin darkening. Both these antagonist responses were both hormone and neuron controlled [95]. It is well known that amphibian and fish melanophores aggregate their melanosomes upon the addition of melatonin, which binds to a specific high-affinity receptor that activates G-proteins to inhibit adenyl cyclase [96,97].

In order to check whether our phytomelatonin-rich extract possessed active functional properties, we studied its ability to aggregate melanophores in fish, since the melanophores contract their granules upon exposure to synthetic melatonin and to phytochemical-rich extract (Bioriex). In both fish species tested, we found very similar structures in the fish caudal fin, melanocytes with a completely disaggregated pigmentation, especially of the fin rays, and melanocytes with partly-to-fully aggregated pigmentation in the areas between the rays (Figure 3, time 0). The addition of 10 µL of 1 µM synthetic melatonin (1 pmol) to fin samples resulted in the aggregation of pigments in those cells with disaggregated pigments (Figure 3, Panel A). Aggregation started after around 4–5 min and took up to 10 min depending on the sample, tissue zone or initial melanophore stage. Regarding the plant extracts, two samples of Bioriex are assayed, both induced the same pattern of pigments aggregation in melanophores with disaggregated pigmentation as synthetic melatonin (Figure 3, Panel B and C; Appendix A). A negative control with 10 µL of distilled water produced no launch pigment aggregation response (data not shown). In the case of Bioriex samples, the total pigment aggregation needed some minutes more (between 960 and 1200 s) than the synthetic melatonin-treated samples (about 500 s), possibly due to differences in phytomelatonin purity. The results obtained in European sea bass caudal fin samples were similar to those obtained for gilthead sea bream (data not shown). The results show that the phytomelatonin contained in Bioriex acts in a similar way to that of synthetic or animal melatonin, since it is really the same molecule. Thus, the Bioriex extract assayed presented excellent biological properties in terms of the particular action of melatonin in fish as a pigment aggregation agent. Additionally, the same fish bioassay can be recommended for characterizing functionally melatonin-rich extracts from different sources such as plants, algae, yeasts, or bacteria, complementing biochemical characterization.

According to a report published by Transparency Market Research entitled “*Melatonin Market for Food & Beverages, Dietary Supplements, Medicine and Other Applications*”, the global demand of synthetic melatonin was valued at USD 504 million in 2012 and is expected to reach USD 1300 million in 2019, the latter involves about 4000 tons of synthetic melatonin [98]. In addition to inducing sleep and alleviating jet lag, melatonin also presents excellent antioxidant properties, strengthen immunological system, and also is an excellent treatment for cancer, as we mentioned above. Increased stress levels in the general population and the legal dietary supplement status of melatonin in USA are the two major driving factors for melatonin consumption. The lack of side effects of the molecule compared to other sleeping pills gives melatonin the advantage for its consumption. However, melatonin is not as freely available to end-users in Europe and Asia Pacific where it is registered as a prescription drug, except in a 2 mg dosage or lesser (only in some European countries). Although the social demography is similar in North America and Europe, the market is considerably limited in Europe due to the regulated status of melatonin. Thus, in the European Community (EC), prolonged release (PR) melatonin (2 mg) is approved for the treatment of insomnia with poor sleep quality in patients aged ≥55 years, being an effective option for the treatment of insomnia. The recommended dose of PR melatonin is 2 mg to be taken 1–2 h before bedtime and after food. The treatment with melatonin for up to 13 weeks is the usual. In general, animal and human studies have documented that the short-term use of melatonin is safe, even in very high doses. In both short-term and long-term clinical trials, melatonin PR has registered excellent results as a remedy for the various parameters of insomnia and sleep quality. No side effects have been found, just some mild effects, such as headaches, nasopharyngitis, back pain, and arthralgia, which in some cases were also felt by the placebo patients. Melatonin is neither physiologically nor psychologically addictive. However, some recommendations concerning the possible risks of melatonin use in specific patient groups were provided [99,100]. 

The use of synthetic melatonin as a dietary supplement represents is the largest application segment, constituting 52.5% of the world demand for melatonin. This segment is also expected to grow a great deal in the coming years. Melatonin supplementation in food and beverages seems to be a new market, which may experience high growth in the immediate future. However, it is still at the expense of FDA authorization. Additionally, melatonin has been used in cosmetic creams. For example, melatonin-based creams significantly improved skin tonicity and skin hydration in women, with a significant reduction in skin roughness, supporting the skin anti-aging effect of this molecule when applied topically [101]. Several cosmetic lines containing synthetic melatonin are marketed, but not containing phytomelatonin.

The assays using phytomelatonin for healthcare purposes are reduced to a handful of experiments in animals, but also in humans. Therefore, in grain-fed chickens, blood melatonin levels can be altered by feeding phytomelatonin rich. The same happened in rats fed with nuts, and also with sprouted beans, which confirms that phytomelatonin is absorbed by the gastrointestinal tract, being eliminated in the urine as 6-sulfatoxymelatonin (aMT6s) [56,102,103,104]. In humans, the consumption of controlled amounts of sweet cherries, plums, or grape juice by young, middle-aged, and elderly subjects led also to an increase in urinary aMT6s [105,106,107,108]. Additionally, the intake of sweet cherries positively influences some sleep-quality parameters, such as sleep efficiency, sleep time, and nocturnal activity in volunteers [109]. The same effects have been observed in tart cherry juice [110]. In a cross-over trial, serum melatonin levels increased up to five times after the consumption of banana, orange, or pineapple, even though these tropical fruits contain very low levels of phytomelatonin [111,112]. In an interesting study, volunteers who consumed between 330 and 660 mL of beer significantly increased their blood melatonin levels compared to a control group. Total antioxidant levels in the blood, which are related to the antioxidant potential of phytomelatonin and other phytocomposites such as flavonoids, also increased [113,114,115]. The protective effects of coffee infusion against liver diseases such as hepatic fibrosis in rats has been related with the antioxidant capacity of the coffee compounds and, in particular, to the high phytomelatonin content in coffee species [116]. Additionally, a recent study suggested that dietary supplementation with melatonin and phytomelatonin should be considered for preventing different disease settings, particular in the liver [117]. As a conclusion, the consumption of plant foods containing phytomelatonin influences blood melatonin and antioxidant levels, which may change certain physiological responses mediated by this indoleamine and also act against diseases.

In the global market, one or two products formulated with phytomelatonin exist. In this regard, note that some manufacturers use the term phytomelatonin in the description of the contents of their products, most of them being products based on synthetic melatonin with, in some cases, some plant extracts added. This is commonly the case with mixed biphasic release formulations, where synthetic melatonin, which is rapidly released at the beginning, is added to an MAP mixture that releases its contents more slowly; it is generally unknown how much phytomelatonin MAP contains.

One commercial product is Sleep Support^®^ (Christchurch, New Zealand) which contains phytomelatonin obtained from freeze-dried Montmorency tart cherry skin extracts, and which is sold in tablets containing 15 μg phytomelatonin/pill, a very low amount to improve sleep quality. The phytomelatonin content of tart cherries is around 14 ng/g fruit [118]. Another better known product is Herbatonin^®^ (Symphony Natural Health, West Valley City, UT, USA), formulated in pills containing 0.3 or 3 mg of phytomelatonin, while in Spain as Zentrum^®^ (Ynsadiet Labs., Leganés, Madrid, Spain) is sold in 1 mg doses. This formulation contain several plant species such as rice (*Oryza sativa* L.) and alfalfa (*Medicago sativa* L.), together with the green alga *Chlorella pyrenoidosa* Chick. These plant species contain very low levels of phytomelatonin: 1 ng/g in rice and 16 ng/g in alfalfa [119]. The presence of Chlorella suggest that the phytomelatonin is mainly obtained by cultivating the green alga in bioreactors, possibly feeding with the phytomelatonin precursor, *L*-tryptophan, as in *Achillea millefolium* L. [120]. Unfortunately, the use of synthetic *L*-tryptophan, if this is the case, would lead to a higher presence of unwanted synthetic by-products, as mentioned above. Another aspect to take into account when using green alga in cultures is the frequent presence of cyanotoxins (microcystins, anatoxin-a, dihydroanatoxin-a, epoxyanatoxin-a, cylindrospermopsin, and saxitoxin, among others) due to contaminations by cyanobacteria (blue-green algae). These cyanotoxins have several unwanted effects related with their possible carcinogenicity, hepatotoxicity, neurotoxicity, cytotoxicity, and dermatotoxicity. Thus, the detection of many cyanotoxins in some algal dietary supplements reinforces the need for a better quality control as regards the potential risks associated with the consumption of these algal supplements [121,122,123]. 

Another interesting product which, as far as we know is not yet available commercially, is yeast-melatonin. Germann and co-workers generated *Saccharomyces cerevisiae* strains that contained heterologous genes encoding several melatonin biosynthesis enzymes and co-factor supporting pathways [124]. The transgenic yeast codified different genes from *Rattus norvegicus, Lactobacillus ruminis, Pseudomonas aeruginosa*, *Homo sapiens*, *Schistosoma mansoni*, *Bos Taurus*, and *Salmonella enterica*. The final strain of *Saccharomyces* cultures were fed with glucose, producing 14.5 mg of melatonin/L, a very substantial amount, in a three-day fermentation cycle. Nevertheless, according to other authors, some improvements in productivity are still necessary to make this process economically interesting for industrial-scale production. For example, problems such as the high *N*-acetylserotonin accumulation in yeast cells, unbalanced gene expression and the identification of some potential toxic intermediates must be addressed [125]. In short, the possible use of transgenic yeast to produce substances for human consumption does not seem a minor problem in the search for a solution to the industrial scale production of melatonin in the near future. 

## 4. Conclusions

In conclusion, based on the herbal mix (HM) and the green-extractive processes describe in this work, stabilized natural phytomelatonin-rich extracts (Bioriex), free of unwanted by-products, were obtained. The final phytomelatonin content of Bioriex was between 5 and 10 mg/g, a sufficient concentration to be used as initial material in interesting applications, such as food supplements, functional foods, cosmetics, and, possibly, in cancer experimentation. The cost-effective industrial-scale production of phytomelatonin is our next goal to satisfy a sector of the market that is increasingly looking towards natural products as an alternative for this relevant molecule with multiple applications.

## Figures and Tables

**Figure 1 antioxidants-09-00158-f001:**
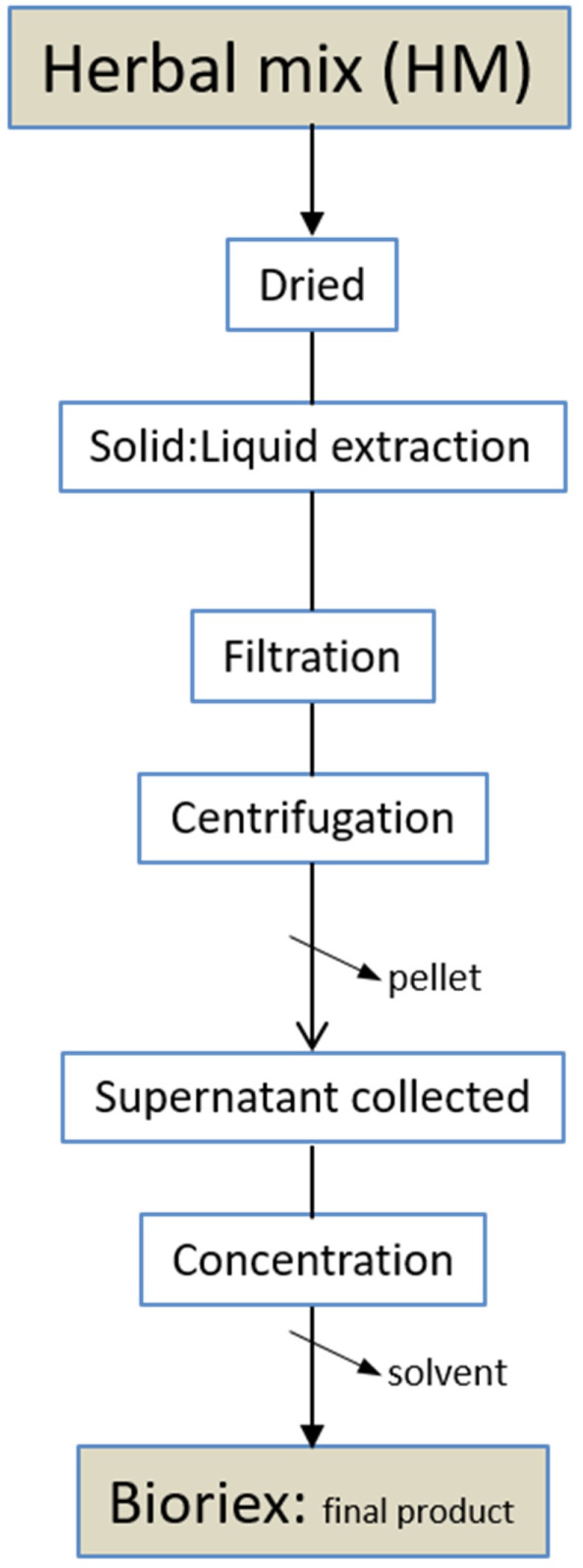
Scheme of the protocol used to obtain phytomelatonin-rich extracts (Bioriex) from herbal mixed selected plants (HM).

**Figure 2 antioxidants-09-00158-f002:**
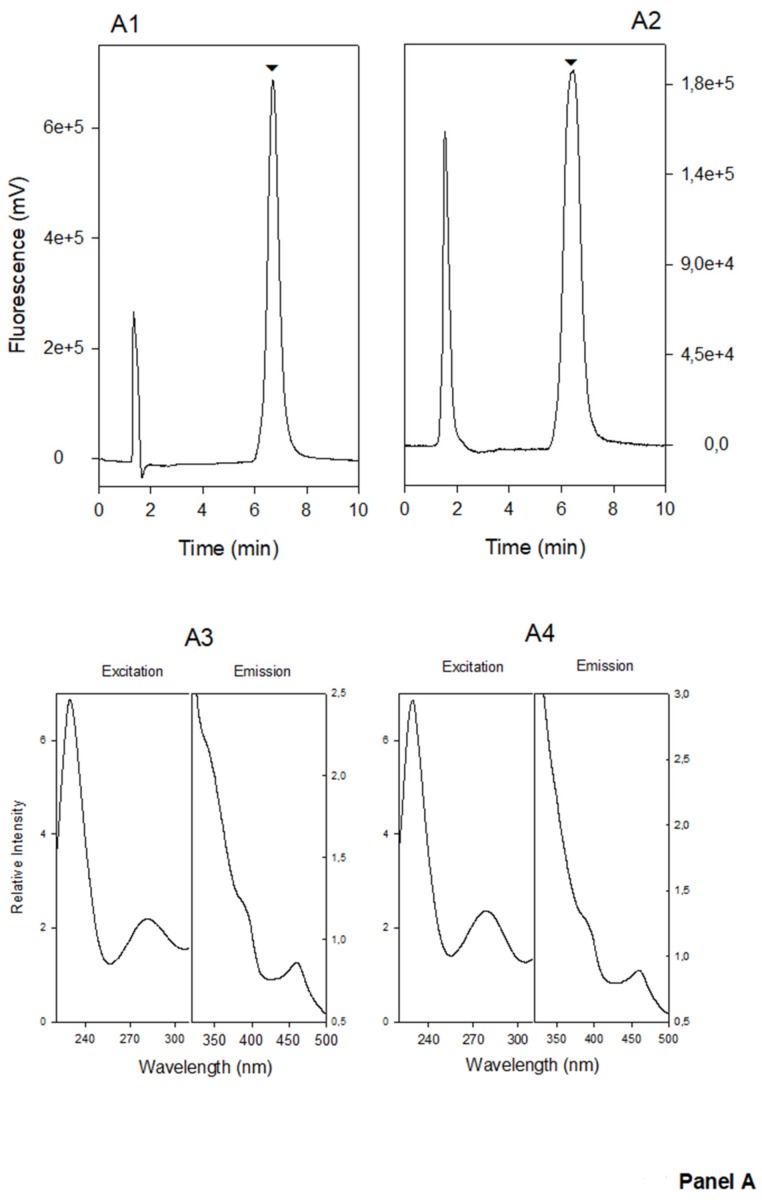
**Panel A**. Representative chromatograms of standard melatonin (**A1**) and Bioriex sample (**A2**) with liquid chromatography and fluorescence detection. Fluorescence detection was programmed at λ_exc_ of 280 nm and λ_emi_ of 350 nm. The melatonin peak has a retention time of 6.5 min. The excitation spectrum was measured with emission at 350 nm and the emission spectrum with excitation at 280 nm, using an in-line spectral analysis (stopped-flow) in standard melatonin (**A3**) and Bioriex sample (**A4**). **Panel B**. Representative chromatograms and mass spectra of standard melatonin, HM and Bioriex samples using liquid chromatography with time-of-flight/mass spectrometry (LC-QTOF/MS). (**B1**): Extracted ion chromatogram (EIC) at *m*/*z* 233.13000 of a standard solution of melatonin. (**B2**): Accurate mass spectra of standard melatonin solution showing different derivates of the base peak ion with the mass data analysis of protonated [M + H]^+^. (**B3**): EIC as in B1 of Bioriex sample. (**B4**): Protonated base peak ion of melatonin in Bioriex sample and their corresponding accurate mass analysis.

**Figure 3 antioxidants-09-00158-f003:**
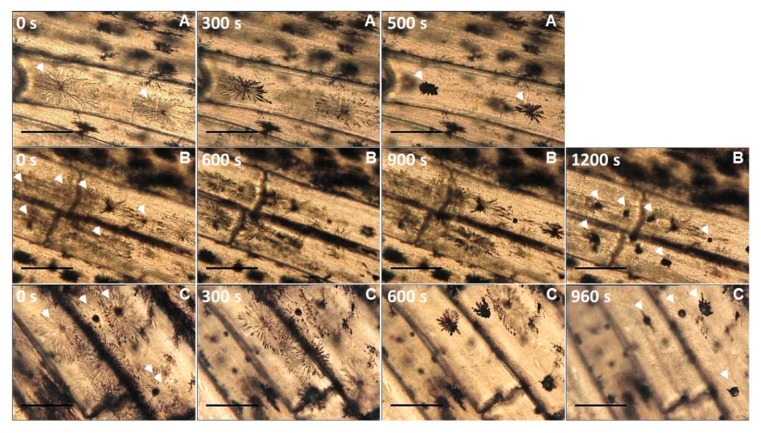
Representative light microscopy images of gilthead seabream fin samples exposed with either synthetic melatonin (**Pael A**), Bioriex-sample-1 (**Pael B**) and Bioriex-sample-2 (**Pael C**) at different times. Bar = 100 µm. Arrowheads indicate melanophores that greatly vary in their morphology from 0 s to the final exposure time.

**Table 1 antioxidants-09-00158-t001:** Common by-products present in synthetic melatonin preparations.

1,3-diphthalimidopropane
hydroxy-bromo-propylphthalimide
Chloro-propylphthalimide
1,2,3,4-tetrahydro-β-carboline-3-carboxylic acid
3-(phenylamino)-alanine (PAA)
1,1′-ethylidenebis-(tryptophan) (so-called *peak E*)
2-(3-indolylmethyl)-tryptophan
formaldehyde-melatonin
formaldehyde-melatonin condensation products
5-hydroxy-tryptamine derivatives
5-methoxy-tryptamine derivatives
*N*-acetyl- and diacetyl-indole derivatives

**Table 2 antioxidants-09-00158-t002:** Proximate analysis of herbal mix (HM) and final product (Bioriex).

Components (in %)	HM Plants	Bioriex
Moisture	93.2 ± 4.3	11.3 ± 0.7
Ash	4.1 ± 0.2	traces
Crude proteins	36.3 ± 1.7	7.1 ± 0.4
Crude fats	5.2 ± 0.3	85.5 ± 4.8
Dietary fibre	20.8 ± 1.1	1.6 ± 0.1
NFEM * (~carbohydrates)	33.6 ± 1.6	5.8 ± 0.4

* NFEM, Nitrogen-free extractive material.

**Table 3 antioxidants-09-00158-t003:** Specific analysis of herbal mix (HM) and final product (Bioriex).

Parameter	HM Plants	Bioriex
Phytomelatonin content	5.5 ± 0.3 µg/g DW	7.2 ± 0.4 mg/g DW
Total phenolic content (TPC) (eq. gallic acid/g DW)	121.8 ± 8.2 nmoles/g DW	126.1 µmoles ± 9.1/g DW
Total flavonoid content (TFC) (eq. quercetin/g DW)	19.8 nmoles ± 0.9/g DW	43.7 µmoles ± 1.9/g DW
Hydrophilic antioxidant activity (HAA) (eq. ascorbic acid/g DW)	61.7 nmoles ± 3.3/g DW	64.2 µmoles ± 3.5/g DW
Total carotenoids (eq. β-carotene/g DW)	0.44 nmoles ± 0.03/g DW	11.92 nmoles ± 0.68/g DW

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
