# Peer review of "Development of a Phytomelatonin-Rich Extract from Cultured Plants with Excellent Biochemical and Functional Properties as an Alternative to Synthetic Melatonin"

_antioxidants, 2020, doi:10.3390/antiox9020158_

Round 1

Reviewer 1 Report

The manuscript submitted by Pérez-Llamas et al. is aimed to characterize a phytomelatonin-rich extract obtained from selected plant material and determine the content in different bioactive compounds. Authors showed that the final phytomelatonin content of the extract was between 5-10 mg/g, which is strikingly high and may find promising applications, specially, in food industry (e.g., natural supplement). Authors also tried to prove the functional properties of the extract by carrying out a bioassay in fish but, in my view, such an election is not completely accurate. Instead, I believe the strength of the manuscript would have benefited from an assay to test the in vivo antioxidant (or sleep-inducing) ability of the phytomelatonin-rich extract.

Author Response

Rev #1

Thank you very much for your comments. Soon we will start a bioassay in animals about the antioxidant capacity of our extract rich in phytomelatonin. Also another bioassay using hamsters to check how our phytomelatonin-rich extract is able to regulate sexual cycles in males. We welcome your suggestions.

Reviewer 2 Report

Arnao and Coworkers presented an interesting phytomelatonin rich extract with very good properties as alternative to synthetic melatonin. The paper is well written and the experiments well designed and detailed. Furthermore the process developed and the biochemical/functional properties make this extract really interesting even for industrial use. Despite the good results a few things can be fixed and improved in the introduction to give a better contextualization of the work. A nice application of melatonin as pro-oxidant and anticancer agent has been developed from Spadoni et al. Chem. Res. Toxicol. 2019, 32, 1, 100-112, this would be a modern example of melatonin activity to be cited. Furthermore the author reports that melatonin synthesis has recently been improved, but the literature on melatonin and related compound synthesis is a bit old and should be integrated with some modern methods, for example Piersanti and coworkers devoted and interesting one step method that can access a variety of tryptamine derivatives including Melatonin (J. Org. Chem. 2015, 80, 6, 3217-3222 and Tetrahedron 2016, 72 (18), 2233-2238), I think that these two references should be added (catalytic green methods with water as byproduct). Beside this I think that the work is really exhaustive and complete, it gives a lot of interesting data on the extract. For the future I suggest to develop a method on HPLC/MS to separate, identify and quantify all the components of the extract and give a correct full characterization, I really think this could be a nice future work.

I Suggest to accept after this minor corrections

Author Response

Rev #2

We welcome all your suggestions. The bibliographic references suggested have been incorporated into new paragraphs in the Introduction section. In the new version, some commentaries on modern melatonin synthesis methods and applications have been incorporated. We welcome your suggestion on a new study by LC-MS of the biomolecules contained in our phytomelatonin-rich extract for its correct and extensive characterization.

Reviewer 3 Report

The study entitled “Development of a phytomelatonin-rich extract from cultured plants with excellent biochemical and functional properties as an alternative to synthetic melatonin” is original, useful, and well-organized. It is evident that the article is written by very experienced scientists. The study is a continuation of previous extensive research of authors in this field. All sections of the manuscript are presented at the desirable scientific level and meet all the formal and content requisites necessary for publication. Authors used established methods, results are interpreted adequately and critically discussed in depth. Conclusions provide adequate replies to the study goals. Manuscript is clear enough and suitable for this prestigious journal. I congratulate the authors to valuable article!

Author Response

Rev #3

Thank you very much for your excellent comments to our work and team.